# Alleviative Effect of Rutin on Zearalenone-Induced Reproductive Toxicity in Male Mice by Preventing Spermatogenic Cell Apoptosis and Modulating Gene Expression in the Hypothalamic–Pituitary–Gonadal Axis

**DOI:** 10.3390/toxins16030121

**Published:** 2024-02-29

**Authors:** Hira Sayed, Qiongqiong Zhang, Yu Tang, Yanan Wang, Yongpeng Guo, Jianyun Zhang, Cheng Ji, Qiugang Ma, Lihong Zhao

**Affiliations:** 1State Key Laboratory of Animal Nutrition and Feeding, Poultry Nutrition and Feed Technology Innovation Team, College of Animal Science and Technology, China Agricultural University, No. 2. West Road Yuanming Yuan, Beijing 100193, China; hirasyed00@gmail.com (H.S.); zqqzqq1996@163.com (Q.Z.); 17801115235@163.com (Y.T.); wyn@cau.edu.cn (Y.W.); jyzhang@cau.edu.cn (J.Z.); jicheng@cau.edu.cn (C.J.); maqiugang@cau.edu.cn (Q.M.); 2College of Animal Science and Technology, Henan Agricultural University, Zhengzhou 450046, China; guoyp@henau.edu.cn

**Keywords:** zearalenone, rutin, mice, testes, reproductive toxicity, antioxidant

## Abstract

Zearalenone (ZEN) is a non-steroidal estrogenic mycotoxin found in many agricultural products and can cause reproductive disorders, mainly affecting spermatogenesis in male animals. Rutin (RUT) is a natural flavonoid compound recognized for its significant antioxidant, anti-inflammatory and estrogenic properties. The present study aimed to determine the protective role of RUT against ZEN-induced reproductive toxicity in male mice. Twenty-four adult Kunming male mice were divided into four groups: control, RUT (500 mg/kg RUT), ZEN (10 mg/kg ZEN), ZEN + RUT (500 mg/kg RUT + 10 mg/kg ZEN), with six replicates per treatment. The results indicated that RUT mitigated ZEN-induced disruption in spermatogenic cell arrangement, decreased spermatozoa count, and increased sperm mortality in the testes. RUT significantly restored ZEN-induced reduction in T, FSH, LH, and E2 serum levels. Moreover, RUT mitigated ZEN-induced apoptosis by increasing the mRNA expression level of *bcl-2*, decreasing the mRNA expression level of *kiss1-r*, and decreasing the protein expression level of caspase 8 in reproductive tissues. These findings indicate the protective role of RUT against ZEN-induced reproductive toxicity in male mice by regulating gonadotropin and testosterone secretions to maintain normal spermatogenesis via the HPG axis, which may provide a new application direction for RUT as a therapeutic agent to mitigate ZEN-induced reproductive toxicity.

## 1. Introduction

Food and feed contaminated with mycotoxins is a worldwide agricultural problem [1]. According to reports from the Food and Agriculture Organisation (FAO) of the United Nations, 25% of crops receive mycotoxin contamination during their growth and storage phases [2]. ZEN is a well-known mycotoxin that shares structural similarities with the important oestrogen hormone 17-beta-estradiol [3]. Due to their structural resemblance, ZEN and reproductive hormones may interfere with their normal functioning. Specifically, ZEN can disrupt luteinizing and follicle-stimulating hormones essential for controlling the hypothalamic–pituitary–gonadal (HPG) axis and ultimately change testosterone levels [4]. In females, persistent exposure to ZEN can significantly affect reproductive health by disturbing the hormonal balance. This disruption may lead to irregular menstrual cycles, ovulatory dysfunction, and reproductive system disorders, potentially increasing the risk of fertility issues and adverse reproductive outcomes [5,6].

Male spermatogenesis can be affected negatively by ZEN’s endocrine-disrupting effects, which may also adversely affect sperm quantity and quality [7]. Moreover, the estrogenic effects of ZEN may cause abnormalities in the male reproductive system, which could result in problems, including testicular atrophy and overall dysfunction in the reproductive system [8,9]. Despite these observed effects, the precise mechanisms underlying ZEN-induced adverse impacts on male reproduction via the HPG axis remain elusive. Therefore, exploring effective strategies to mitigate ZEN-induced reproductive damage in animal health is essential.

Many studies have indicated the potential of various compounds such as lycopene [10], selenium [9], vitamin C [11], curcumin [12], procyanidins [13], and glucosamine [14] in preventing ZEN-induced reproductive toxicities. Among these, rutin (RUT), recognized as rutinoside, quercetin-3-O-rutin, is a prevalent flavonoid found abundantly in fruits and vegetables. Studies have shown that RUT exhibits a range of pharmacological actions, including antibacterial, hepatoprotective, immunomodulatory, antioxidant, and estrogenic properties [15,16]. According to previous studies, RUT protects against reproductive damage caused by chemicals such as streptozotocin [17], cyclophosphamide [18], cadmium [19], cisplatin [20], and busulfan [21]. Furthermore, research has shown that RUT decreases cisplatin-induced renal injury and cell death by reducing TNF-α, NF-κB, and mRNA expression levels of *caspase-3* in wister rats [22]. A study by Kandemir et al. [23] found that RUT reduces apoptosis and oxidative stress to mitigate gentamicin-induced nephrotoxicity. Moreover, Elsawy et al. [24] conducted a study indicating that RUT protects male mice from hepatorenal toxicity and hypogonadism caused by carbon tetrachloride (CCl4). The observed effect was related to activating antioxidant enzymes, specifically superoxide dismutase and glutathione peroxidase and reduced lipid peroxidation levels in testes tissues.

Despite these compelling findings, the precise effects of RUT on ZEN-induced reproductive damage and the underlying mechanism are still unknown. Therefore, the present study aims to comprehensively investigate RUT’s potential ameliorative effects and action mechanisms against ZEN-induced reproductive toxicity in male mice.

## 2. Results

### 2.1. Effects of RUT on the Testes Tissue Morphology and Sperm Quality of Mice Fed with ZEN

The potential protective effects of RUT against ZEN exposure were investigated through histological analysis using H&E staining on testicular tissue. The findings depicted in Figure 1A–D shows distinct observations. In the control group (Figure 1A1,A2) and RUT group (Figure 1B1,B2), the testes of mice exhibited a tightly connected spermatogenic epithelium with the presence of spermatogenic cells across all stages. The sequential arrangement of the spermatogenic epithelium with distinct layers and a central lumen filled with sperm (red star) was notably prominent in the RUT group compared to the control group. Conversely, the ZEN-exposed group (Figure 1C1,C2) displayed significant alterations: a discernible cavity (indicated by the red arrow) along the seminiferous tubule, a decrease in spermatozoa count (illustrated by the black curved arrow), and a disorganized arrangement of spermatogenic cells at all stages (indicated by the black straight arrow). However, the ZEN + RUT group (Figure 1D1,D2) exhibited notable mitigation of these detrimental effects.

Additionally, quantitative results in Figure 1E–G showed that exposure to ZEN resulted in a significant decrease (*p* < 0.05) in the testicular organ index and an increase (*p* < 0.05) in sperm deformity. On the other hand, when RUT was added to the diet, the testicular organ index showed a significant increase (*p* < 0.001). There was an interaction effect (*p* < 0.01) on sperm mortality between ZEN exposure and RUT addition. The sperm mortality rate was significantly higher (*p* < 0.01) than the CON group in the ZEN-exposed group; however, this trend was significantly reversed (*p* < 0.001) by RUT supplementation in the ZEN + RUT group.

### 2.2. Effects of RUT on Serum Hormones, the mRNA Expression Levels of Reproductive Hormone Receptor and Steroid Synthesis-Related Genes in Testes of Mice Fed with ZEN

As shown in Figure 2A–D, dietary RUT levels and ZEN exposure had notable interactive effects on the serum levels of E2 (*p* < 0.001), T (*p* < 0.0001), FSH (*p* < 0.0001), and LH (*p* < 0.0001). Subsequent multiple comparison analysis indicated that the levels of E2, T, FSH, and LH in serum from the ZEN-exposed group exhibited a significant decrease compared to those from the control group. Notably, adding RUT to the diet substantially reversed serum E2 and LH changes induced by ZEN in mice (ZEN + RUT). We also studied the mRNA expression levels of reproductive hormone receptors and steroid synthesis-related genes through RT-qPCR. The mRNA expression levels of the reproductive hormone receptor (*ER-α*, *ER-β*, *LH-R*, *FSH-R)* are shown in Figure 2E–H, which indicated that ZEN exposure caused a significant decrease in the mRNA expression level of *ER-α* (*p* < 0.05) in testes. Furthermore, ZEN significantly decreased the mRNA expression level of *CYP11-A* (*p* < 0.05) in the testicular tissue of mice, as shown in Figure 2I–K. These findings suggest that ZEN induces alterations in serum hormones and the mRNA expression levels of reproductive hormone receptors and genes associated with steroid synthesis in male mice, and dietary RUT supplementation mitigates these alterations to some extent.

### 2.3. Effect of RUT on the Cell Apoptosis and the mRNA and Protein Expression Levels of Genes Related to Apoptosis in the Testes of Mice Fed with ZEN

To explore the protective effect of RUT against ZEN-induced cell apoptosis, we employed the in situ terminal transferase labeling (TUNEL) technique to assess cell apoptosis within the testes of mice. As seen in Figure 3A, spermatogenic cells at all stages in spermatogenic tubules of the testes showed varying degrees of apoptosis in the ZEN group compared to the control group. Conversely, there was a significant decrease in spermatogenic cell death in the spermatogenic tubules of testes in the RUT+ZEN group. The quantitative analysis through RT-qPCR further confirmed the study results, as shown in Figure 3B–D. We found a significant interaction between dietary RUT and ZEN exposure on the mRNA expression level of *bcl-2* (*p* < 0.05) in the testes of mice. The findings of multiple comparisons showed that, in comparison to the control group, ZEN treatment reduced the mRNA expression level of *bcl-2* in the mice testes. However, compared to the ZEN group, the mRNA expression level of *bcl-2* in the testes was eliminated in the ZEN + RUT group. Furthermore, testicular tissue from the ZEN group showed a significant increase in the mRNA expression level of *caspase-8* (*p* < 0.05) compared to the CON group, and the addition of RUT in the diet reversed these changes induced by ZEN. Western blot results indicated that ZEN and RUT treatment had no significant effect on the expression levels of Bcl-2 and caspase-3 proteins while ZEN significantly increased the protein expression level of caspase-8 *(*Figure 3E–H). These findings suggest that RUT inhibits the apoptotic signaling cascade by improving the structural integrity of the spermatogenic tubules in testes of mice exposure ZEN.

### 2.4. Effects of RUT on the mRNA Expression Levels of Reproductive Hormone Receptors and Apoptosis-Related Genes in the Hypothalamus of Male Mice Fed with ZEN

We also evaluated mRNA expression levels of reproductive hormone receptors and apoptosis-related genes to explore the effect of RUT on ZEN-induced toxicity in the hypothalamus (Figure 4A–H). ZEN exposure caused a significant decrease in the mRNA expression level of *ER-β* in the hypothalamus (*p* < 0.05) as compared to the RUT + ZEN group.

### 2.5. Effects of RUT on the mRNA Expression Levels of Reproductive-Related Genes in the Pituitary Gland of Male Mice Fed with ZEN

The pituitary gland of male mice exposed to ZEN experienced a significant decrease in mRNA expression level of *GnRH-R* (*p* < 0.01), as illustrated in Figure 5A,B. Furthermore, there was an interaction between dietary RUT level and ZEN exposure on the mRNA expression level of *kiss1-r* (*p* < 0.05) in the pituitary gland of male mice. The results of multiple comparisons indicated that, as compared to the control group, the ZEN group had a significant increase in the mRNA expression level of *kiss1-r* in the pituitary gland of mice. However, this effect was notably reversed in the ZEN + RUT group.

## 3. Discussion

Zearalenone, a prominent estrogenic mycotoxin primarily produced by fungi, poses a global public health threat due to its pervasive presence in animal feeds [25]. In male mice, ZEN profoundly affects the reproductive system by disrupting the HPG axis, leading to hormone regulation, impaired spermatogenesis, and altered reproductive signaling pathways [26,27,28]. Flavonoids, known for their antioxidant and anti-inflammatory properties, have shown potential in mitigating reproductive toxicity induced by ZEN [29,30]. Rutin, a flavonoid compound, has antioxidant, anti-inflammatory, immunomodulatory, and regulating reproductive hormone properties [30,31]. In general, RUT’s structural similarity to estradiol plays an essential role in alleviating the toxicity caused by ZEN. RUT has the potential to competitively bind with estrogen receptors, indicating that it could mitigate the adverse effects of ZEN on these receptors. This action could contribute to regulating the disturbed hormonal balance induced by ZEN exposure, especially within the reproductive system.

Our study aimed to evaluate the protective effects of RUT against ZEN-induced reproductive damage in male mice. Initially, we assessed various reproductive parameters, revealing significant alterations in male mice exposed to ZEN: decreased testes organ index, increased sperm mortality, and deformity rates. Microscopic examination of the testicular seminiferous epithelial tissue in ZEN-exposed mice depicted structural abnormalities such as loose, hollow, reduced sperm count within seminiferous tubules and irregular arrangement of germ cells, indicative of ZEN-induced damage to testicular integrity. Recent reports by Li et al. [32] suggested that ZEN could influence spermatogenesis through ferroptosis, potentially caused by iron accumulation in testicular tissue, leading to irregular germ cell arrangements in the testes. However, pretreatment with RUT exhibited notable differences in the reproductive epithelium, showing increased structural integrity and more uniform cell arrangements. In male mice, similar protective effects of RUT against testicular toxicity caused by various chemicals have been reported [21,24]. ZEN-induced reproductive toxicity significantly decreased serum T, LH, FSH, and E2 levels. This decrease correlates with increased lipid peroxidation due to free radicals, which affects the leydig cells, the brain, and the HPG axis [7,33]. Our results showed reduced estrogen levels because ZEN can compete with 17β-estradiol for binding the cytosol estrogen receptor, reducing estrogen production and causing endocrine disruption. Abarikwu et al. [19] highlighted the antioxidant potential of RUT in alleviating cadmium-induced reproductive toxicity in mice, while Elsawy et al. [24] studied RUT’s ability to restore decreased serum T, LH, and FSH levels in male mice exposed to CCl4 due to its antioxidant properties. Our study aligns with these findings, indicating that RUT effectively ameliorated the ZEN-induced endocrine disturbances.

ZEN significantly decreased mRNA expression levels of *ER-α* in testes tissues while mRNA expression levels of *ER-β* unchanged because ZEN significantly influences estrogen signaling pathways by acting as a competitive *ER-α* agonist and an *ER-β* antagonist [34]. Notably, RUT supplementation elevated the mRNA expression levels of these genes, indicating its potential to mitigate ZEN-induced ER stress in testicular tissue through antioxidative and free radical scavenging capabilities.

Furthermore, ZEN influences crucial gonadal receptors such as *FSH-R* and *LH-R*, which are pivotal for steroidogenesis and gametogenesis in mice [35]. The *LH-R* is located on the testicular membrane, serving as the exclusive binding site for LH. It activates the cyclic adenosine monophosphate (cAMP)/protein kinase A (*PKA*) signaling pathway on binding to *LH-R* [36]. The substrate *PKA* activates *StAR*, which facilitates the transfer of cholesterol into mitochondria and essential transcription factors that control the expression of genes related to steroidogenesis [37]. Cholesterol is converted to testosterone through various enzymes, including *CYP11-A* and *3β-HSD*, in multiple steps [38]. In this study, the mRNA expression level of *FSH-R* was not changed in the testes of ZEN-treated mice, possibly due to self-regulation mechanisms in response to ZEN metabolism.

However, the mRNA expression level of steroidogenic gene *CYP11-A* in the testes was downregulated in the ZEN group compared with that in the control group. This disruption in steroidogenic gene expression aligns with previous reports on the effect of ZEN on the mRNA expression levels of testicular *LH-R*, *StAR*, *CYP11-A*, and *3β-HSD* in male mice [38]. Surprisingly, RUT mitigated ZEN’s detrimental effect on steroidogenesis, possibly by scavenging ROS produced by ZEN, which may impact the transcription and protein levels of steroidogenic enzyme activity [39]. ZEN-induced apoptosis, believed to result from oxidative stress, involves the upregulation of *caspase-3* and *caspase-8* and the downregulation of *bcl-2* within testicular tissue [40,41]. In our study, we noted that ZEN exposure downregulated the mRNA expression level of *bcl-2* and upregulated the mRNA expression level of *caspase-8* and its protein expression level within testicular tissue. This indicates that several signaling pathways, such as those connected to the *bcl-2* family, endoplasmic reticulum stress, caspase-dependent processes, and intrinsic mitochondrial pathways, may enhance apoptosis in testicular cells by ZEN [42,43]. The administration of RUT attenuates apoptosis induced by ZEN in testicular cells, likely attributed to its capability to scavenge ROS, thereby mitigating oxidative stress-induced apoptosis.

The hypothalamus–pituitary–gonadal axis (HPG) is an essential component of the neuroendocrine system responsible for regulating the development of mammalian reproductive organs and controlling reproductive functions. Endocrine system disruptors, like ZEN, can interfere with the HPG axis and negatively affect the reproductive system [44]. Kisspeptin binds to its corresponding receptor, *kiss1-r*, enabling the hypothalamus to produce more gonadotropin-releasing hormone (*GnRH*). *GnRH-R* receptor of the pituitary gland is activated by *GnRH*, producing and secreting LH and FSH. In turn, LH and FSH bind to the corresponding receptors on the testes, *LH-R* and *FSH-R*, to control gonadotropin production [45,46].

Furthermore, RUT and ZEN exhibited a notable interaction with the mRNA expression level of *kiss1-r* in the pituitary gland of mice. A study revealed that treatment with 5 μg/L ZEN led to a notable increase in the mRNA expression level of *ER-α* in the brain, gonads, and liver [47]. This could potentially lead to contradictions, as our study revealed an increase in the hypothalamus’s mRNA expression level of *ER-β*. The differences in test results were possibly caused by the different species and ZEN concentrations used in the respective studies. Interestingly, pretreatment with RUT mitigated the reproductive toxicity induced by ZEN through modulation of the HPG axis in male mice (Figure 6).

## 4. Conclusions

In summary, RUT might alleviate ZEN-induced reproductive toxicity by modulating the expression levels of genes related to reproductive function, restoring hormone balance by regulating the HPG axis, preserving testicular structure integrity, ameliorating apoptosis, and protecting spermatogenesis. These findings highlight RUT as a potential therapeutic agent against ZEN-induced reproductive toxicity.

## 5. Materials and Methods

### 5.1. Chemicals

RUT was purchased from Shanghai Aladdin Health Chemical Technology Co., Ltd., Shanghai, China, and ZEN standards were purchased from Qingdao Pribolab Engineering Co., Ltd., Qingdao, China. All chemicals used in the research were of analytical grade.

### 5.2. Experimental Animals and Design

The experimental procedures conducted in this study received approval from the institutional animal care committee of China Agricultural University (AW21111202-1-5) Kunming male mice were purchased from SPF Biotechnology Co., Ltd. (Beijing, China) and placed in a controlled environment (24 °C, alternating 12 h light/dark cycles) throughout the experiment.

Following an adaptive rearing phase, 24 (5 weeks old) mice were divided into four groups (*n* = 6 per group): the control group (basic diet), the RUT group (basic diet + 500 mg/kg RUT), ZEN group (basic diet + 10 mg/kg ZEN); and ZEN+RUT group (basic diet + 500 mg/kg RUT + 10 mg/kg ZEN). ZEN and RUT were individually dissolved in 40 mL of chromatographic-grade methanol and 40 mL of ultrapure water. Subsequently, the resultant solutions were evenly sprayed onto the daily feed, and allowed to evaporate overnight at room temperature. The feed was then uniformly mixed, water was added, and the mixture was stirred at a 1000:480 feed-to-water ratio. Subsequently, the mixture was formed into cylindrical pellets with a 1.5 cm diameter using a pellet mill. The pellets were baked for 5–6 h at 50 °C. The experimental duration spanned 28 days. The basic diet used in this study was purchased from Beijing Huafukang Biotechnology Co., Ltd., Beijing, China.

### 5.3. Collection of Sample

On the last day of the experiment, a necropsy was carried out, and the final body weight of the mice was recorded. Blood samples were obtained using the eyeball extraction and decapitation method, and subsequently centrifuged for 20 min at 3000 rpm. The extracted upper serum was kept at −20 °C in sterile centrifuge tubes.

Subsequently, the cranial cavity of the mice was carefully opened to extract the hypothalamus and pituitary glands. These tissues were preserved in liquid nitrogen. The testes and epididymis were removed from the abdominal cavity, quickly cleaned of adherent connective tissues, and weighed to calculate the organ coefficient (organ weight divided by body weight). One side of the testes was placed in 4% paraformaldehyde fixative for pathological observation, and other lateral testes sections were preserved in cryopreservation tubes and immediately submerged in liquid nitrogen to assess sperm quality.

### 5.4. Determination of Mouse Sperm Quality

Sperm mortality: To create a sperm suspension, one epididymis was added to 1 mL of normal saline. The sperm were then activated by incubating the mixture at 37 °C for 15 min. After incubation, 10 μL of the sperm suspension was dropwise-spread onto a preheated counting chamber or slide. Sperm mortality was graded using WHO criteria; the number of inactive/dead sperm in five squares was recorded as Grade IV. The whole process was completed within 10 min to ensure accuracy. The sperm mortality rate (%) was calculated as follows:$$Sperm\;Mortality\;rate\;\left(\%\right)\;=\frac{Number\;of\;inactive\;sperm\;(Grad\;IV)\;}{Total\;number\;of\;observed\;sperm\;(M)}\;\times \;100$$

Sperm deformity: A total of 100 μL of sperm suspension was mixed with 10 μL of 2% eosin stain and allowed to stand for 5–10 min for proper staining. A drop of stained sperm suspension was placed on a glass slide, fixed with methanol, and allowed to air dry to produce a smear. Deformed sperm were observed using a high-power lens under a microscope. A total of 200 complete spermatozoa from each mouse were systematically examined for abnormalities, and the number (m) of total abnormal spermatozoa among these 200 spermatozoa was calculated. The sperm deformity rate (%) was calculated as follows:$$Sperm\;deformity\;rate\;\left(\%\right)\;=\;\frac{m}{Mouse\;total\;sperm\;count\;\left(200\right)}\;\times \;100\%$$

### 5.5. Reproductive Hormone Indicators of Serum

The concentrations of E2, T, FSH, and LH in the blood serum were detected according to the kit’s instructions, and the kits were all bought from Shanghai Yuanmu Biotechnology Co., Ltd. (Shanghai, China).

### 5.6. Histopathological Examinations of Testes

For the histopathological examination, testes were fixed in 4% paraformaldehyde solution for 24 h, dried in ethanol, cleaned with xylene, and embedded in paraffin wax. Tissue sections of 5 μm thickness were precisely cut and dyed with hematoxylin–eosin (H&E) dye. The slides were studied using an Olympus BX61 light microscope (Olympus, Tokyo, Japan) to examine the histopathological changes. An Olympus DP70 digital camera (Olympus, Japan) was used to take the images. Three slices from each treatment were observed through various magnifications to ensure a comprehensive assessment of histopathological alterations and detailed observations.

### 5.7. Observation of Testis Cell Apoptosis via the In Situ Terminal Transferase Labeling Method (TUNEL Method)

The paraffin-embedded sections of testes specimens were subjected to TUNEL staining using a Fluorescein (FITC) Tunel Cell Apoptosis Detection Kit (Wuhan Xavier Biotechnology Co., Ltd., Wuhan, China). Briefly, sections underwent deparaffinization and dehydration using graded concentrations of xylene and ethanol.

Subsequently, proteinase K digestion was then completed in 25 min at room temperature. This was followed by washing in PBS and incubating the TUNEL reaction mixture (consisting of the labeling and enzyme solution) for 2 h at 37 °C in a humid environment. Negative controls were prepared following the same procedure, excluding the incubation step with TdT. Identification of TUNEL-positive cells relied on intense dark labeling, equivalent to or exceeding that observed in apoptotic cells in the simultaneously labeled positive control slide. A fluorescence microscope facilitated the observation of TUNEL-stained sections, and photographic documentation was conducted. Each treatment group was represented by three slices and examined at various magnifications.

### 5.8. Quantitative Real-Time PCR (RT-qPCR) to Determine the mRNA Expression Levels of Genes Related to the Reproductive Hormone, Steroid Synthesis, and Cell Apoptosis

Total RNA extraction from mouse testis samples was conducted using the FastPure^®^ Cell/Tissue Total RNA Isolation Kit V2 (RC112; Vazyme Biotech Co., Ltd., Nanjing, China) following the manufacturer’s protocol. The extracted RNA quality and quantity were evaluated spectrophotometrically using the 260/280 nm absorbance ratio with a NanoDrop-2000 spectrophotometer (ThermoFisher Scientific Co., Waltham, MA, USA). Reverse transcription of the isolated RNA into complementary DNA (cDNA) was performed by using the HiScript^®^ II Q RT SuperMix for qPCR (+gDNA wiper) (R223-01; Vazyme Biotech Co., Ltd., Nanjing, China) according to the manufacturer’s protocol. The resultant cDNA was kept for further examination at −20 °C. Taq Pro Universal SYBR qPCR Master Mix (Q712-02; Vazyme Biotech Co., Ltd., Nanjing, China) was used for two-step quantitative real-time PCR on RT-qPCR detection systems (Bio-Rad, Hercules, CA, USA, CFX ConnectTM) according to the manufacturer’s instructions. Gene-specific primer pairs associated with reproduction were designed using GenBank sequences obtained from the National Center for Biotechnology Information and are detailed in Table 1. The following PCR conditions were applied to each sample: initial denaturation at 95 °C for 5 min, followed by 40 cycles of denaturation at 95 °C for 10 s, annealing at 58 °C for 30 s, and extension at 72 °C for 5 min. Each sample was analyzed in triplicate. Melting curve analysis was used to determine the specificity of the PCR product. Each target gene’s relative mRNA expression levels were determined by comparing them to the expression of the housekeeping gene, GAPDH, using the 2^−ΔΔCT^ method [48]. The values derived from the control group served as the calibrator for comparative analysis.

### 5.9. Western Blot Analysis

The 20 mg samples with the RIPA buffer containing protease inhibitors (PMSF) were homogenized properly to ensure proper protein extraction. Protein concentration was determined using the BCA protein assay method, following the manufacturer’s instructions. The extracted proteins were then resolved on 10% SDS-PAGE and transferred to the PVDF membrane. The membrane was blocked with 5% non-fat milk in TBS-T for 1.5 h at room temperature and washed with TBS-T. Primary antibodies against *Bcl-2*, *Caspase-3*, and *Caspase-8*, diluted according to the manufacturer’s recommendations, were applied to the membrane and incubated for 2 h on a shaker. The membrane was washed three times for 10 min each with TBS-T to remove unbound primary antibodies. The membrane was incubated with secondary antibodies, specifically goat anti-rabbit IgG (H+L)-HRP conjugate, diluted according to the manufacturer’s recommendations, for 1 h at room temperature on a shaker. The protein bands were visualized using an enhanced chemiluminescence reagent, clarity Western ECL Substrate (Bio-Rad, CA, USA). GADPH was used as a loading control to ensure equal loading of proteins across the lanes. Band intensities were quantified using image analysis software (Image Pro-Plus 6.0, Media Cybernetics, Silver Spring, MD, USA). Normalization of *Bcl-2*, *caspase-3*, and *caspase-8* band intensities to the loading control (GADPH) was performed.

### 5.10. Statistical Analysis

Data was analysed using GraphPad Prism V 8.0.1 (GraphPad Software, San Diego, CA, USA). As a 2 × 2 factorial arrangement, two-way ANOVA was used to determine the main effects of ZEN exposure and dietary RUT level, and their interaction, Duncan’s multiple comparison was used to separate means when interactive effects significantly different. Results are presented as the means ± SEMs. All statements of significance were based on *p* < 0.05.

## Figures and Tables

**Figure 1 toxins-16-00121-f001:**
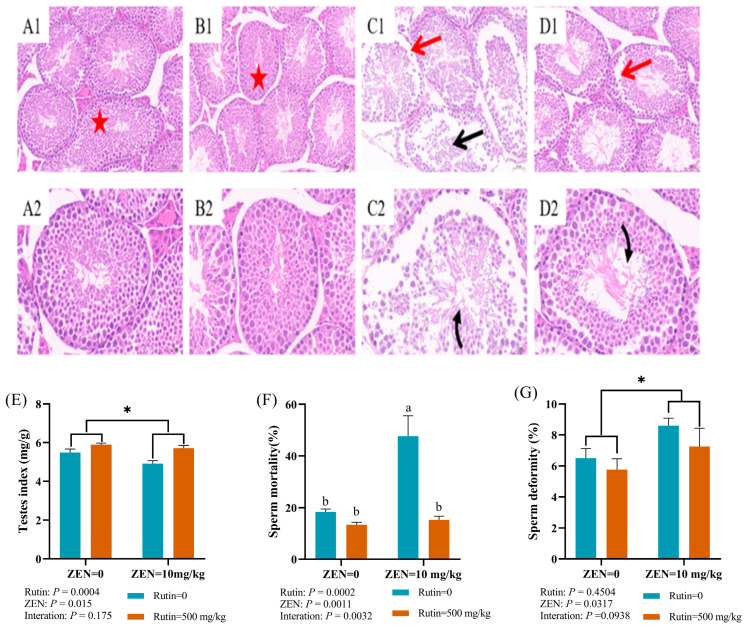
Effect of RUT on the testes tissue morphology and sperm quality of mice fed with ZEN. Note: (**A**–**D**) represent the hematoxylin and eosin (H&E) staining results of testicular morphology in the control, RUT, ZEN, and ZEN + RUT groups, respectively. (**A1**–**D1**) and (**A2**–**D2**) represents the 20× and 40× views, respectively, and the scale bars are 50 micrometres (μm) and 20 μm, respectively. Red star indicate healthy immature and maturing spermatozoa, red straight arrows denote the testicular seminiferous epithelial tissue displaying loose connections and a visible cavity, black curved arrows represent a reduced number of spermatozoa, and black straight arrows indicate degenerated or non-functional sperm cells. The effects of RUT on (**E**) testicular organ index, (**F**) sperm mortality, and (**G**) sperm deformity in mice after ZEN treatment are shown (mean ± standard error, *n* = 6). ^a, b^ values without the same letters were significantly different (*p* < 0.05). * significant main effect (*p* < 0.05) of ZEN exposure.

**Figure 2 toxins-16-00121-f002:**
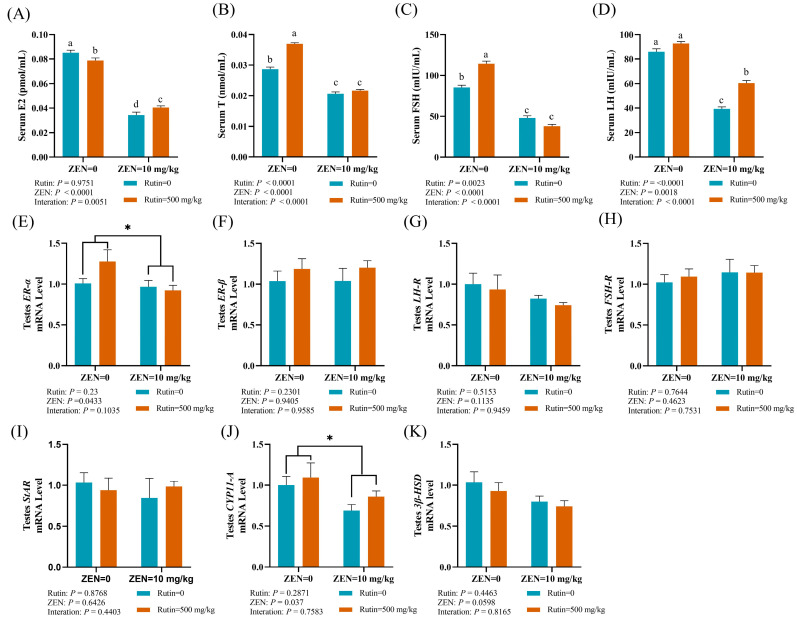
Effects of RUT on serum hormones: (**A**) estradiol (E2), (**B**) testosterone (T), (**C**) follicle-stimulating hormone (FSH), and (**D**) luteinizing hormone (LH). The mRNA expression levels of reproductive hormone receptors: (**E**) estrogen receptor-alpha (*ER-α*), (**F**) estrogen receptor-beta (*ER-β*), (**G**) luteinizing hormone receptor (*LH-R*), and (**H**) follicle-stimulating hormone receptor (*FSH-R*). The expression levels of steroid synthesis-related genes: (**I**) steroidogenic acute regulatory protein (*StAR*), (**J**) cytochrome P450 family 11 subfamily A member 1 (*CYP11-A*), and (K) 3-beta-hydroxysteroid dehydrogenase (*3β-HSD*) in the testes of mice after ZEN treatment are shown (mean ± standard error, *n* = 6). ^a, b, c, d^ values without the same letters were significantly different (*p* < 0.05). * significant main effect (*p* < 0.05) of ZEN exposure.

**Figure 3 toxins-16-00121-f003:**
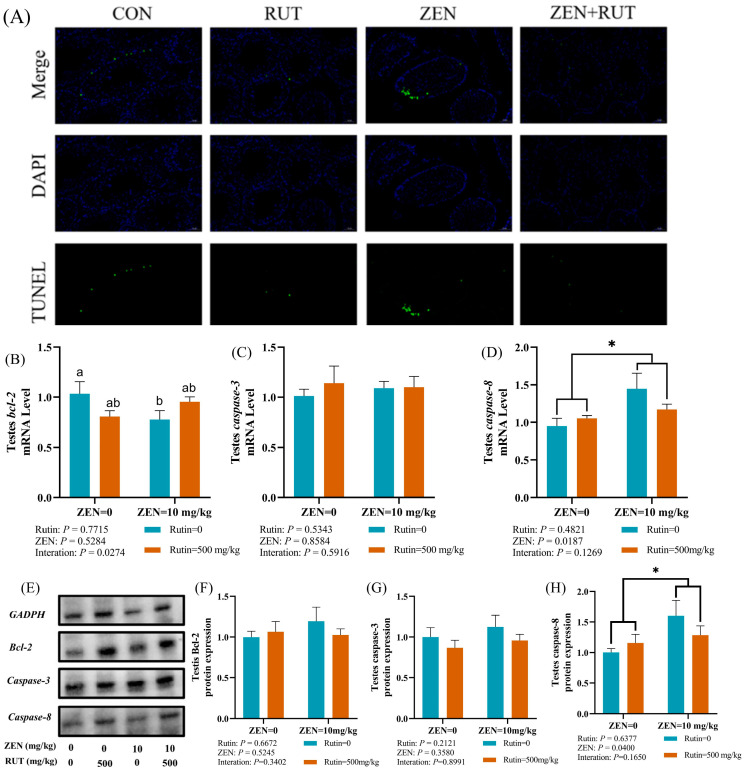
Effects of RUT on cell apoptosis and mRNA expression levels of genes related to apoptosis: (**A**) Terminal deoxynucleotidyl transferase dUTP nick end labeling (TUNEL) staining in the testicular tissue sections of mice in each treatment group; green represents TUNEL-positive cells, blue represents nuclei, and Merge represents the overlay of the two markers. The scale bar is 50 micrometres (μm), mRNA expression levels of *B-cell lymphoma 2* (*bcl-2*), *caspase-3*, *caspase-8* (**B**–**D**), and protein expression level of Bcl-2, caspase-3, caspase-8 (**E**–**H**) in the testes of mice after ZEN treatment (mean ± standard error, *n* = 6). ^a, b^ values without the same letters were significantly different (*p* < 0.05). * significant main effect (*p* < 0.05) of ZEN exposure.

**Figure 4 toxins-16-00121-f004:**
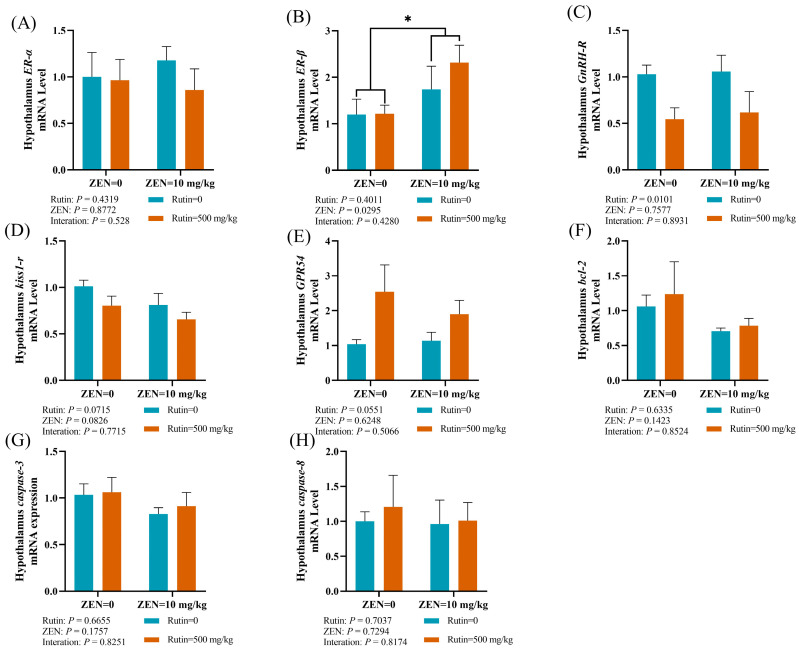
Effects of RUT on the mRNA expression levels of reproductive hormone receptors: (**A**) estrogen receptor-alpha (*ER-α*), (**B**) estrogen receptor-beta (*ER-β*), (**C**) gonadotropin-releasing hormone receptor (*GnRH-R*), (**D**) kisspeptin receptor (*kiss1-r*), and (**E**) G protein-coupled receptor 54 (*GPR54*), as well as apoptosis-related genes: (**F**) *B-cell lymphoma 2* (*bcl-2*), (**G**) *caspase-3*, and (**H**) *caspase-8* in the hypothalamus of mice after ZEN treatment (mean ± standard error, *n* = 6). * significant main effect (*p* < 0.05) of ZEN exposure.

**Figure 5 toxins-16-00121-f005:**
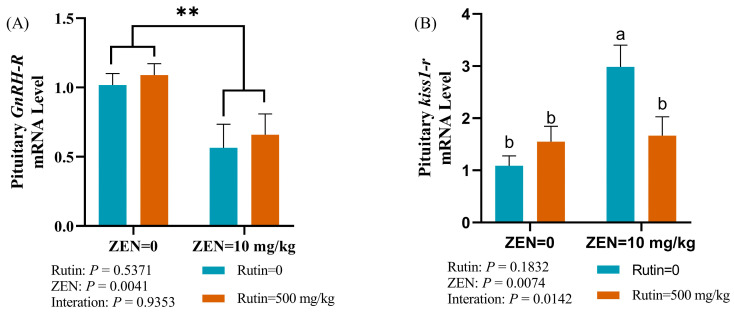
Effects of RUT on the mRNA expression levels of reproductive-related genes: (**A**) gonadotropin-releasing hormone receptor (*GnRH-R*) and (**B**) kisspeptin receptor (*Kiss1-r*) in the pituitary gland of mice after ZEN treatment (mean ± standard error, *n* = 6). Bars bearing different superscripts are significantly different from each other (*p* < 0.05). ^a, b^ values without the same letters were significantly different (*p* < 0.05). ** significant main effect (*p* < 0.01) of ZEN exposure.

**Figure 6 toxins-16-00121-f006:**
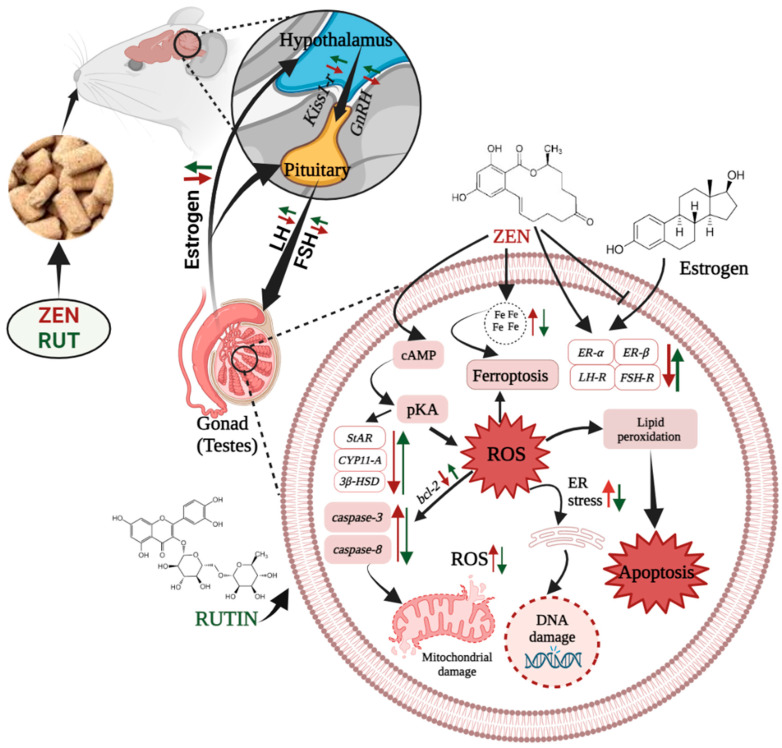
The protective mechanism of RUT against ZEN-induced reproductive damage in male mice.

**Table 1 toxins-16-00121-t001:** The quantitative real-time PCR primers used in the study.

Gene	Forward Primer (5′-3′)	Reverse Primer (3′-5′)
*ER-α*	TCTGGAGTGTGCCTGGTTGGAG	GCGGAATCGACTTGACGTAGCC
*ER-β*	ACGTCAGGCACATCAGTAACAAGG	CATCTCCAGCAGCAGGTCATACAC
*LH-R*	TCTTGGAAATGCTACACAGCAAC	GGAGGAGGGCAAAATACACAAA
*FSH-R*	CAAGATAGCAAGGTGACCGAGA	GCAAGYYGGGYAGGYYGGAGA
*StAR*	GGGCATACTCAACAACCAGGAA	CTTGACATTTGGGTTCCACTCTC
*CYP11-A*	CATTCCTGCTGGAAACTGTGAGC	ATCTCGCTGAGCTTTCTTGTAGG
*3β-HSD*	AGCAAAAAGATGGCCGAGAA	GGCACAAGTATGCAATGTGCC
*Bcl-2*	TGAAGCGGTCCGGTGGATA	CAGCATTTGCAGAAGTCCTGTGA
*Caspase-3*	AGAGACATTCATGGGCCTGAAATAC	CACCATGGCTTAGAATCACACACAC
*Caspase-8*	TGGGACCTGCTGGTCAACTTC	AAAGATCTCAATTCCAACTCGCTCA
*GnRH-R*	CTGCCTTCAATGCTTCCTTC	CAGTGGCATGACGATCAGAG
*Kiss1-r*	TCTGGGCTCACTGCATGTCCTAC	CTGTCTGAAGTGTGAACCCAGGAAG
*GAPDH*	AGGTCGGTGTGAACGGATTTG	GGGGTCGTTGATGGCAACA

## Data Availability

The raw data supporting the conclusions of this article will be made available by the authors on request.

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
