# Peer review of "Alleviative Effect of Rutin on Zearalenone-Induced Reproductive Toxicity in Male Mice by Preventing Spermatogenic Cell Apoptosis and Modulating Gene Expression in the Hypothalamic–Pituitary–Gonadal Axis"

_toxins, 2024, doi:10.3390/toxins16030121_

Round 1
Reviewer 1 Report
Comments and Suggestions for Authors
This paper is a well- documented paper with supporting experiments. I have only two comments.
Please explain how did you chose the dose of rutin? Is it possible to achive 500 mg/kg rut in normal diet?
Please, add the limitations of your study
Besides, the main question addressed by the research is whether Rutin has a protective role against zearalenone-induced damage in male mice
The article examined a natural compound that has not yet been tested for its ability to neutralise the toxic effects of ZEA in the male reproductive system.
This paper expands knowledge about RUT in response to the action of ZEA
I think that it would be nice to check also GPER1 expression since it may be associated with the action of ZEA (doi: 10.1038/s41598-021-86788-w). Some Western Blot experiments (for example caspases, bax or bcl2) should also be performed because the protein expression level is also very important.
I think that based on the presented results we couldn’t clearly state whether RUT has or not protective effect
In my opinion, the references are appropriate
The tables are fine. Expansion of abbreviations should appear in figure descriptions.
Reviewer 2 Report
Comments and Suggestions for Authors
I consider that it is positive both the originality and the results and discussions correlated with the conclusions regarding the mode of action of rutin on zearalenone-induced reproductive toxicity by preventing apoptosis and modulating gene expression related to reproductive hormones.
I consider that the article can be published after a few corrections regarding the English language
Comments on the Quality of English LanguageMinor editing of English required
Reviewer 3 Report
Comments and Suggestions for Authors
This is an interesting article describing the beneficial effects of rutin against zearalenone toxicity in male mice. The methodological approach seems correct, and the results are of scientific interest. However, the authors should indicate the rationale for rutin level used in the study. In addition, some improvements in the text are required, as follows:
-Title: Should be summarized, and without abbreviations.
-Abstract: Some abbreviations remain in the text.
-Figure 3: Resolution of this figure should be improved.
-All figures: Do not use "ppm", use only international standard units (e.g., mg/mk).
-Section 5.2: How was ZEN incorporated in the diet? Was the target level determined by any analytical method? This is important for accepting the validity of data described in the experiment.
Comments on the Quality of English LanguageThe English throughout the text is good. Some expression, such as "The worldwide agricultural problem is the contamination of food and feed with mycotoxins", may require some improvement depending on the language style of the journal.
